# Factor Graph Neural Network

**Zhen Zhang**[1]    **Fan Wu** [2]    **Wee Sun Lee**[3]

[1] Australian Institute for Machine Learning & The University of Adelaide, Australia
[2] University of Illinois at Urbana-Champaign
[3] School of Computing, National University of Singapore

zhen.zhang02@adelaide.edu.au    fanw6@illinois.edu    leews@comp.nus.edu.sg

## Abstract

Most of the successful deep neural network architectures are structured, often consisting of elements like convolutional neural networks and gated recurrent neural networks. Recently, graph neural networks (GNNs) have been successfully applied to graph-structured data such as point cloud and molecular data. These networks often only consider pairwise dependencies, as they operate on a graph structure. We generalize the GNN into a factor graph neural network (FGNN) providing a simple way to incorporate dependencies among multiple variables. We show that FGNN is able to represent Max-Product belief propagation, an approximate inference method on probabilistic graphical models, providing a theoretical understanding on the capabilities of FGNN and related GNNs. Experiments on synthetic and real datasets demonstrate the potential of the proposed architecture.

## 1   Introduction

Deep neural networks are powerful approximators that have been extremely successful in practice. While fully connected networks are universal approximators, successful networks in practice tend to be structured, *e.g.*, grid-structured convolutional neural networks and chain-structured gated recurrent neural networks (*e.g.*, LSTM, GRU). Graph neural networks [7, 34, 35] have recently been successfully used with graph-structured data to capture pairwise dependencies between variables and to propagate the information to the entire graph.

The dependencies in the real-world are often beyond pairwise connections. *E.g.*, in the LDPC encoding, the bits of a signal are grouped into several clusters. In each cluster, the sum of all bits should be equal to zero [36]. Then in the decoding procedure, these constraints should be respected. In this paper, we show that the GNN can be naturally extended to capture dependencies over multiple variables by using the factor graph structure. A factor graph is a bipartite graph with a set of variable nodes connected to a set of factor nodes; each factor node indicates the presence of dependencies among its connected variables. We call a neural network formed from the factor graph a factor graph neural network (FGNN).

Factor graphs have been used extensively to specify Probabilistic Graph Models (PGMs) for modeling dependencies among multiple random variables. In PGMs, the specification or learning of the model is usually separate from the inference process. Approximate inference algorithms such as Belief Propagation which is often used, since inference over PGMs are often NP-hard. Unlike PGMs, graph neural networks usually learn a set of latent variables and the inference procedure at the same time in an end-to-end manner; the graph structure only provides information on the dependencies along which information propagates. For problems where domain knowledge is weak, or where approximate inference algorithms do

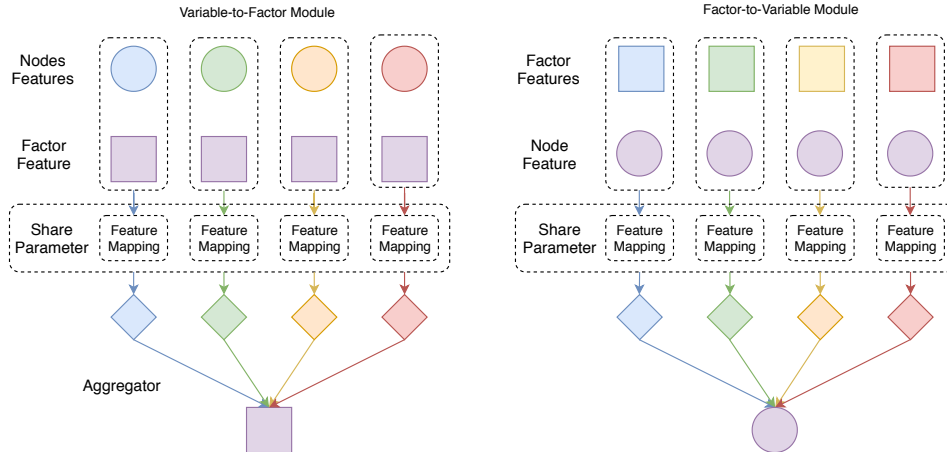

Figure 1: The structure of the Factor Graph Neural Network (FGNN): the Variable-to-Factor (VF) module is shown on the left while the Factor-to-Variable (FV) module is shown on the right.

poorly, being able to learn an inference algorithm jointly with the latent variables, specifically for the target data distribution, often produces superior results.

We take the approach of jointly learning the inference algorithm and latent variables in developing the factor graph neural network (FGNN). The FGNN is defined using two types of modules, the Variable-to-Factor (VF) module and the Factor-to-Variable (FV) module (see Figure 1). These modules are combined into a layer, and the layers are stacked together into an algorithm. We show that the FGNN is able to exactly parameterize the Max-Product Belief Propagation, which is a widely used approximate *maximum a posteriori* (MAP) inference algorithm for PGMs. Theoretically, this shows that FGNN is at least as powerful as Max-Product and hence can solve problems solvable by Max-Product, *e.g.*, [2, 11].

The simplest representation of PGMs uses a tabular potential for the factors. Unfortunately, its size grows exponentially with the number of variables in the factors, which makes higher order tabular factors impractical. We design FGNN to naturally allow approximation of the factors by parameterizing factors in terms of the maximum of a set of rank-1 tensors. The parameterization can represent any factor exactly with a large enough set of rank-1 tensors; the number of rank-1 tensors required can grow exponentially for some problems but may be small for easier problems. Using this representation, the size of the FGNN that can simulate Max-Product grows polynomially with the number of rank-1 tensors in approximating the factors, giving a practical approximation scheme that can be learned from data.

The theoretical relationship with Max-Product provides understanding on the representational capabilities of GNNs in general, and of FGNN in particular. From the practical perspective, the factor graph provides a flexible way for specifying dependencies. Furthermore, inference algorithms for many types of graphs, *e.g.*, graphs with typed edges or nodes, are easily developed using the factor graph representation. Edges, or more generally factors, can be typed by tying together parameters of factors of the same type, or can also be conditioned from input features by making the edge or factor parameters a function of the features; nodes can similarly have types or features with the use of factors that depend on a node variable. With typed or conditioned factors, the factor graph can also be assembled dynamically for each graph instance. FGNN provides a flexible learnable architecture for exploiting these graphical structures – just as factor graph allows easy specification of different types of PGMs, FGNN allows easy specification of both typed and conditioned variables and dependencies as well as a corresponding data-dependent approximate inference algorithm.

To be practically useful, the FGNN architecture needs to be practically *learnable* without being trapped in poor local minimums. We performed experiments to explore the practical potential of FGNN. FGNN performed well on a synthetic PGM inference problem with constraints on the number of elements that may be present in subsets of variables. We also experimented with applying FGNN on the LDPC decoding and long term human motion

prediction. We outperform the standard LDPC decoding method under some noise conditions and achieve state-of-the-art results on human motion prediction, demonstrating the potential of the architecture.

## 2 Background

Probabilistic Graph Models (PGMs) use graphs to model dependencies among random variables. These dependencies are conveniently represented using a factor graph, which is a bipartite graph $\mathcal{G} = (\mathcal{V}, \mathcal{C}, \mathcal{E})$ where each vertex $i \in \mathcal{V}$ in the graph is associated with a random variable $x_i \in X$, each vertex $c \in \mathcal{C}$ is associated with a function $f_c$ and an edge connects a variable vertex $i$ to factor vertex $c$ if $f_c$ depends on $x_i$.

Let $\mathbf{x}$ be the set of all variables and let $\mathbf{x}_c$ be the subset of variables that $f_c$ depends on. Denote the set of indices of variables in $\mathbf{x}_c$ by $s(c)$. We consider discrete PGM as follows

$$p(\mathbf{x}) = \frac{1}{Z} \exp \left[ \sum_{c \in \mathcal{C}} \theta_c(\mathbf{x}_c) + \sum_{i \in \mathcal{V}} \theta_i(x_i) \right], \quad (1)$$

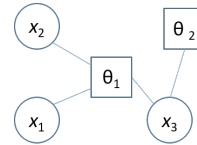

Figure 2: A factor graph where $f_1$ depends on $x_1$, $x_2$, and $x_3$ while $f_2$ depends on $x_3$.

where $\exp(\theta_c(\cdot))$, $\exp(\theta_i(\cdot))$ are positive functions called potentials (with $\theta_c(\cdot)$, $\theta_i(\cdot)$ as the corresponding log-potentials) and $Z$ is a normalizing constant. The goal of *maximum a posteriori* (MAP) inference [16] is to find the assignment which maximizes $p(\mathbf{x})$, that is

$$\mathbf{x}^* = \operatorname*{argmax}_{\mathbf{x}} \sum_{c \in \mathcal{C}} \theta_c(\mathbf{x}_c) + \sum_{i \in \mathcal{V}} \theta_i(x_i). \quad (2)$$

As Eq. (2) is NP-hard in general [29], approximation are often required. One common method is Max-Product Belief Propagation, which is an iterative method formulated as

$$b_i(\mathbf{x}_i) = \theta_i(x_i) + \sum_{c:i \in s(c)} m_{c \to i}(x_i); \quad m_{c \to i}(x_i) = \max_{\hat{\mathbf{x}}_c : \hat{x}_i = x_i} \left[ \theta_c(\hat{\mathbf{x}}_c) + \sum_{i' \in s(c), i' \neq i} b_{i'}(\hat{x}_{i'}) \right]. \quad (3)$$

Max-product type algorithms are fairly effective in practice, achieving moderate accuracy in various problems [6, 8, 32].

**Related Works** Various graph neural network models have been proposed for graph structured data, including methods based on the graph Laplacian [3, 4, 13], gated networks [18], and various other neural networks structures for updating the information [1, 5, 9, 28]. Gilmer et al. [7] show that these methods can be viewed as applying message passing on pairwise graphs and are special cases of Message Passing Neural Networks (MPNNs).

In this work, we seek to go beyond pairwise interactions by using message passing on factor graphs. Recent works on the expressive power of graph neural networks have also consider using higher order networks, e.g. Morris et al. [25] and Maron et al. [21] consider networks based on higher order Weisfeiler-Lehman tests that can be used for testing graph isomorphism. In contrast to graph isomorphism, FGNN builds on probabilistic graphical models, which provide a rich language allowing the designer to specify prior knowledge in the form of pairwise as well as higher order dependencies in a factor graph.

## 3 Factor Graph Neural Network

Previous works on graph neural networks focus on learning pairwise information exchanges. The Message Passing Neural Network (MPNN) [7] provides a framework for deriving different graph neural network algorithms by modifying the message passing operations. We aim at enabling the network to efficiently encode higher order features and to propagate information between higher order factors and the nodes by performing message passing on a factor graph. We describe the FGNN network and show that for specific settings of the network parameters we obtain the Max-Product Belief Propagation algorithm for the corresponding factor graph.

## 3.1 Factor Graph Neural Network

First we give a brief introduction to the Message Passing Neural Network (MPNN), and then we propose one MPNN architecture which can be easily extended to a factor graph version. Given a graph $\mathcal{G} = (\mathcal{V}, \mathcal{N})$, where $\mathcal{V}$ is the node set and $\mathcal{N}$ is the adjacency list, assume that each node is associated with a feature $\mathbf{f}_i$ and each edge $(i, j) : i \in \mathcal{V}, j \in \mathcal{N}(i)$ is associated with an edge feature $\mathbf{e}_{ij}$. Then a message passing neural network layer is defined in [7] as

$$\mathbf{m}_i = \sum_{j \in \mathcal{N}(i)} \mathcal{M}(\mathbf{f}_i, \mathbf{f}_j, \mathbf{e}_{ij}), \qquad \tilde{\mathbf{f}}_i = \mathcal{U}_t(\mathbf{f}_i, \mathbf{m}_i), \tag{4}$$

where $\mathcal{M}$ and $\mathcal{U}$ are usually parameterized by neural networks. The summation in (4) can be replaced with other aggregator, *e.g.*, maximization [31]. The main reason to use maximization is that summation may be corrupted by a single outlier, while maximization is more robust. Thus in our paper we also use the maximization as aggregator. There are also multiple choices of the architecture of $\mathcal{M}$ and $\mathcal{U}$. We propose an MPNN architecture as follows

$$\tilde{\mathbf{f}}_i = \max_{j \in \mathcal{N}(i)} \mathcal{Q}(\mathbf{e}_{ij}) \mathcal{M}(\mathbf{f}_i, \mathbf{f}_j), \tag{5}$$

where $\mathcal{M}$ maps feature vectors to a length-$n$ feature vector, and $\mathcal{Q}(\mathbf{e}_{ij})$ maps $\mathbf{e}_{ij}$ to a $m \times n$ matrix. Then by matrix multiplication and aggregation a new length-$m$ feature is generated.

---

**Algorithm 1** The FGNN layer

**Input:** $\mathcal{G}(\mathcal{V}, \mathcal{C}, \mathcal{E}), [\mathbf{f}_i]_{i \in \mathcal{V}}, [\mathbf{g}_c]_{c \in \mathcal{C}}, [t_{ci}]_{(c,i) \in \mathcal{E}}$

**Output:** $[\tilde{\mathbf{f}}_i]_{i \in \mathcal{V}}, [\tilde{\mathbf{g}}_c]_{c \in \mathcal{C}}$

1: **Variable-to-Factor:**

2: $\quad \tilde{\mathbf{g}}_c = \max_{i:(c,i) \in \mathcal{E}} \mathcal{Q}(\mathbf{t}_{ci} | \Phi_{\mathrm{VF}}) \mathcal{M}([\mathbf{g}_c, f_i] | \Theta_{\mathrm{VF}})$

3: **Factor-to-Variable:**

4: $\quad \tilde{f}_i = \max_{c:(c,i) \in \mathcal{E}} \mathcal{Q}(\mathbf{t}_{ci} | \Phi_{\mathrm{FV}}) \mathcal{M}([\mathbf{g}_c, f_i] | \Theta_{\mathrm{FV}})$

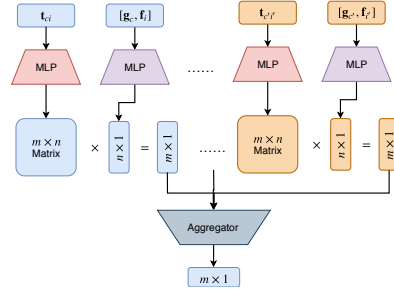

Figure 3: **Left:** The pseudo code for the FGNN layer. Here the Variable-to-Factor (VF) module and the Factor-to-Variable (FV) modules are MPNN layers with similar structure but different parameters. **Right:** The detailed architecture for our VF or FV module.

The MPNN encodes unary and pairwise edge features, but higher order features are not directly encoded. Thus we extend the MPNN by introducing extra factor nodes. Given a factor graph $\mathcal{G} = (\mathcal{V}, \mathcal{C}, \mathcal{E})$, unary features $[\mathbf{f}_i]_{i \in \mathcal{V}}$ and factor features $[\mathbf{g}_c]_{c \in \mathcal{C}}$, assume that for each edge $(c, i) \in \mathcal{E}$, with $c \in \mathcal{C}, i \in \mathcal{V}$, there is an associated edge feature vector $[t_{ci}]$. Then, the Factor Graph Neural Network layer on $\mathcal{G}$ can be extended from (5) as shown in Figure 3 and Algorithm 1, where $[\Phi_{\mathrm{VF}}, \Theta_{\mathrm{VF}}]$ are parameters for the Variable-to-Factor module, and $[\Phi_{\mathrm{FV}}, \Theta_{\mathrm{FV}}]$ are parameters for the Factor-to-Variable module.

We use the same architecture for sending the messages from variables to factors as well as for the messages from factors to variables. If the aim is only to simulate the Max-Product algorithm, it would be more direct to have different architectures for the two types of messages. However, having the same architecture is simpler to implement. In addition, it is also possible to have a variant where messages from variables and factors are sent simultaneously instead of alternately; in this case we simply have a MPNN on a bipartite (factor) graph with the same structure for the variable and factor nodes.

## 3.2 FGNN for Max-Product Belief Propagation

In this section, we prove that a widely used approximate inference algorithm, Max-Product Belief Propagation can be exactly parameterized by the FGNN. The sketch of the proof is as follows. First we show that any higher order potentials can be decomposed as maximization

over a set of rank-1 tensors, and that the decomposition can be represented by a FGNN layer. After the decomposition, a single Max-Product iteration only requires two operations: (1) maximization over rows or columns of a matrix, and (2) summation over a group of features. We show that the two operations can be exactly parameterized by the FGNN and that $k$ Max-Product iterations can be simulated using $\mathcal{O}(k)$ FGNN layers.

In general, the size of a potential grows exponentially with the number of variables that it depends on. In that case the size of FGNN may explode. However, if the potential can be well approximated as a moderate number of rank-1 tensors, the corresponding FGNN will also be of moderate size. In practice, the potential functions may be unknown and only features of the of the factor nodes are provided; FGNN can learn the approximation from data, potentially exploiting regularities such as low rank approximations if they exist.

**Tensor Decomposition** For discrete variables $x_1, \ldots, x_n$, a rank-1 tensor is a product of univariate functions of the variables $\prod_{i=1}^{n} \phi_i(x_i)$. A tensor can always be decomposed as a finite sum of rank-1 tensors [15]. This has been used to represent potential functions, e.g. in [33], in conjunction with sum-product type inference algorithms. For max-product type algorithms, a decomposition as a maximum of a finite number of rank-1 tensors is more appropriate. It has been shown that there is always a finite decomposition of this type [14].

**Lemma 1** ([14]). *Given an arbitrary potential function $\phi_c(\mathbf{x}_c)$, there exists a variable $z_c \in \mathcal{Z}_c$ with $|\mathcal{Z}_c| < \infty$ and a set of univariate potentials $\{\phi_{ic}(x_i, z_c)|i \in c\}$, s.t.*

$$\log \phi_c(\mathbf{x}_c) = \log \max_{z_c} \prod_{i \in s(c)} \phi_{ic}(x_i, z_c) = \max_{z_c} \sum_{i \in s(c)} \varphi_{ic}(x_i, z_c), \tag{6}$$

*where $\varphi_{ic}(x_i, z_c) = \log \phi_{ic}(x_i, z_c)$.*

Using ideas from [14], we show that a PGM can be converted into single layer FGNN with the non-unary potentials represented as a finite number of rank-1 tensors.

**Proposition 2.** *A factor graph $\mathcal{G} = (\mathcal{V}, \mathcal{C}, \mathcal{E})$ with variable log potentials $\theta_i(x_i)$ and factor log potentials $\varphi_c(\mathbf{x}_c)$ can be converted to a factor graph $\mathcal{G}'$ with the same variable potentials and the decomposed log-potentials $\varphi_{ic}(x_i, z_c)$ using a one-layer FGNN.*

The proof of Proposition 2 and the following propositions can be found in the supplementary material. With the decomposed higher order potential, one iteration of the Max-Product (3) can be rewritten using the following two equations:

$$b_{c \to i}(z_c) = \sum_{i' \in s(c), i' \neq i} \max_{x'_i} \left[ \varphi_{i'c}(x_{i'}, z_c) + b_{i'}(x_{i'}) \right], \tag{7a}$$

$$b_i(x_i) = \theta_i(x_i) + \sum_{c:i \in s(c)} \max_z [\varphi_{ic}(x_i, z_c) + b_{c \to i}(z_c)]. \tag{7b}$$

Given the log potentials represented as a set of rank-1 tensors at each factor node, we show that each iteration of the Max-Product message passing update can be represented by a Variable-to-Factor (VF) layer and a Factor-to-Variable (FV) layer, forming a FGNN layer, followed by a linear layer (that can be absorbed into the VF layer for the next iteration).

With decomposed log-potentials, belief propagation mainly requires two operations: (1) maximization over rows or columns of a matrix; (2) summation over a group of features. We first show that the maximization operation in (7a) (producing max-marginals) can be done using neural networks that can be implemented by the $\mathcal{M}$ units in the VF layer.

**Proposition 3.** *For arbitrary feature matrix $\mathbf{X} \in \mathbb{R}^{m \times n}$ with $x_{ij}$ as its entry in the $i^{th}$ row and $j^{th}$ column, the feature mapping operation $\hat{\mathbf{x}} = [\max_j x_{ij}]_{i=1}^m$ can be exactly parameterized with a $2\log_2 n$-layer neural network with at most $\mathcal{O}(n^2 \log_2 n)$ parameters.*

Following the maximization operations, Eq. (7a) requires summation of a group of features. However, the VF layer uses max instead of sum operators to aggregate features. Assuming that the $\mathcal{M}$ operator has performed the maximization component of equation (7a) producing max-marginals, Proposition 4 shows how the $\mathcal{Q}$ layer can be used to produce a matrix $\mathbf{W}$ that converts the max-marginals into an intermediate form to be used with the max

aggregators. The output of the max aggregators can then be transformed with a linear layer ($\mathbf{Q}$ in Proposition 4) to complete the computation of the summation operation required in equation (7a). Hence, equation (7a) can be implemented using the VF layer together with a linear layer that can be absorbed in the $\mathcal{M}$ operator of the following FV layer.

**Proposition 4.** *For arbitrary non-negative valued feature matrix $\mathbf{X} \in \mathbb{R}_{\geq 0}^{m \times n}$ with $x_{ij}$ as its entry in the $i^{th}$ row and $j^{th}$ column, there exists a constant tensor $\mathbf{W} \in \mathbb{R}^{m \times n \times mn}$ that can be used to transform $\mathbf{X}$ into an intermediate representation $y_{ik} = \sum_{ij} x_{ij} w_{ijk}$, such that after maximization operations are done to obtain $\hat{y}_k = \max_i y_{ik}$, we can use another constant matrix $\mathbf{Q} \in \mathbb{R}^{n \times mn}$ to obtain $[\sum_i x_{ij}]_{j=1}^{n} = \mathbf{Q}[\hat{y}_k]_{k=1}^{mn}$.*

Eq. (7b) can be implemented in the same way as (7a) by the FV layer. First the max operations are done by the $\mathcal{M}$ units to obtain max-marginals. The max-marginals are then transformed into an intermediate form using the $\mathcal{Q}$ units which are further transformed by the max aggregators. An additional linear layer is then sufficient to complete the summation operation required in (7b). The final linear layer can be absorbed into the next FGNN layer, or as an additional linear layer in the network in the case of the final Max-Product iteration.

Using the above two proposition, we can implement all important operations (7). Firstly, by Proposition 3, we can construct the Variable-to-Factor module using the following proposition.

**Proposition 5.** *The operation in (7a) can be parameterized by one MPNN layer with $\mathcal{O}(|X| \max_{c \in \mathcal{C}} |\mathcal{Z}_c|)$ parameters followed by a $\mathcal{O}(\log_2 |X|)$-layer neural network with at most $\mathcal{O}(|X|^2 \log_2 |X|)$ hidden units.*

Meanwhile, by Proposition 3 and Proposition 4 the Factor-to-Variable module can be constructed using the following proposition.

**Proposition 6.** *The operation in (7b) can be parameterized by one MPNN layer, where the $\mathcal{Q}$ network is identity mapping and the $\mathcal{M}$ network consists of a $\mathcal{O}(\max_{c \in \mathcal{C}} \log_2 |\mathcal{Z}_c|)$-layer neural network with at most $\mathcal{O}(\max_{c \in \mathcal{C}} |\mathcal{Z}_c|^2 \log_2 |\mathcal{Z}_c|)$ parameters and a linear layer with $\mathcal{O}(\max_{c \in \mathcal{C}} |c|^2 |X|^2)$ parameters.*

Using the above two proposition, we achieves the main theory result in this paper as follows.

**Corollary 7.** *The max-product algorithm in (3) can be exactly parameterized by the FGNN, where the number of parameters are polynomial w.r.t $|X|$, $\max_{c \in \mathcal{C}} |\mathcal{Z}_c|$ and $\max_{c \in \mathcal{C}} |c|$.*

## 4    Experiments

In this section, we evaluate the models constructed using FGNN for three types of tasks: MAP inference over higher order PGMs, LDPC decoding and human motion prediction.

### 4.1    MAP Inference over PGMs

**Data**   We construct four synthetic datasets (D1, D2, D3 and D4) for this experiment. All models start with a length-30 chain structure with binary-states nodes with node potentials randomly sampled from $\mathcal{U}[0,1]$, and the pairwise potentials encourage two adjacent nodes to take state 1, *i.e.*, it gives high value to configuration $(1,1)$ and low value to others. In D1, the pairwise potentials are fixed, while in the others, they are randomly generated. For D1, D2 and D3, a budget higher order potential [23] is added at every node; these potentials allow at most $k$ of the 8 variables within their scope to take the state 1; specifically, we set $k = 5$ in D1 and D2 and set randomly in D3. In D4, there is no higher order potential at all.

In this paper, we use the simplest, but possibly most flexible method of defining factors in FGNN: we condition the factors on the input features. Specifically, for the problems in this section, all parameters that are not fixed are provided as input factor features. We test the ability of the proposed model to find the MAP solutions, and compare the results with MPNN [7] as well as several MAP inference solver, namely AD3 [23] which solves a linear programming relaxation using subgradient methods, Max-Product Belief Propagation [32], implemented by [24], and a convergent version of Max-Product – MPLP [8], also based on a linear programming relaxation. The approximate inference algorithms are run with the

Table 1: Results (percentage agreement with MAP and standard error) on synthetic datasets with runtime in microseconds in bracket (exact followed by approximate inference runtime for AD3).

| | AD3 | Max-Product | MPLP | MPNN | Ours |
|---|---|---|---|---|---|
| D1 | 80.7±0.0014 (5 / 5) | 57.3±0.0020 (6) | 65.8±0.0071 (57) | 71.9±0.0009 (131) | **92.5**±0.0012 (144) |
| D2 | 83.8±0.0014 (532 / 325) | 50.5±0.0053 (1228) | 68.5±0.0074 (55) | 74.3±0.0009 (131) | **89.1**±0.0010 (341) |
| D3 | 88.1±0.0006 (91092 / 1059) | 53.5±0.0081 (4041) | 64.2±0.0056 (55) | 82.1±0.0008 (121) | **93.2**±0.0006 (382) |
| D4 | **100** (6 / 5) | **100** (6) | 99.9±0.0005 (0.04) | 91.2±0.0005 (137) | 98.0±0.0003 (216) |

correct models while the graph neural network models use learned models, trained with exact MAP solutions generated by a branch-and-bound solver that uses AD3 for bounding [23].

**Architecture and training details** In this task, we use a factor graph neural network consisting of 8 FGNN layers (the details is provided in the supplementary file). The model is implemented using pytorch [27], trained with Adam optimizer [12] with initial learning rate $\text{lr} = 3 \times 10^{-3}$ and after each epoch, lr is decreased by a factor of 0.98. All the models in Table 1, were trained for 50 epoches after which all models achieve convergence.

**Results** The percentage of agreement with MAP solutions is shown in Table 1. Our model achieves far better result on D1, D2 and D3 than all others. D4 consists of chain models, where Max-Product works optimally [1]. The linear programming relaxations also perform well. In this case, our method is able to learn a near-optimal inference algorithm.

Traditional method including Max-Product, MPLP perform poorly on D1, D2 and D3. In these even though FGNN can emulate traditional Max-Product, it is better to learn a different inference algorithm. AD3 have better performance than others, but worse than our FGNN. The accuracy of FGNN is noticeably higher than that of MPNN as MPNN does not use the higher order structural priors that are captured by FGNN.

We also did a small ablation study on the size of the FGNN high order potentials (HOPs) using D1 and D2. On D1, the accuracies are 81.7 and 89.9 when 4 and 6 variables are used in place of the correct 8 variables; on D2, the accuracies are 50.7 and 88.9 respectively. In both cases, the highest accuracies are achieved when the size of the HOPs are set correctly.

## 4.2 LDPC Decoding

The low-density parity check (LDPC) codes is widely used in wired and wireless communication, where the decoding can be done by sum/max-product belief propagation [36].

**Data** Let **x** be the 48-bit original signal, and **y** be the 96-bit LDPC encoded signal by encoding scheme "96.3.963"[19]. Then a noisy signal $\tilde{\mathbf{y}}$ is obtained by transferring **y** through a channel with Gaussian and burst noise, that is, for each bit $i$, $\tilde{y}_i = y_i + n_i + p_i z_i$, where $n_i \sim \mathcal{N}(0, \sigma^2)$ , $z_i \sim \mathcal{N}(0, \sigma_b^2)$, and $p_i$ is a bernoulli random variable *s.t.* $P(p_i = 1) = \eta$; $P(p_i = 0) = 1 - \eta$. In the experiment, we set $\eta = 0.05$ as [10] to simulate unexpected burst noise during transmission. By tuning $\sigma$, we can get different signal with $\text{SNR}_{dB} = 20 \log_{10}(1/\sigma)$.

In the experiment, for all learning based methods, we generate $\tilde{\mathbf{y}}$ from randomly sampled **x** on the fly with $\text{SNR}_{dB} \in \{0, 1, 2, 3, 4\}$ and $\sigma_b \in \{0, 1, 2, 3, 4, 5\}$. For each learning based method, $10^8$ samples are generated for training. Meanwhile, for each different $\text{SNR}_{dB}$ and $\sigma_b$, 1000 samples are generated for validating the performance of trained model.

In LDPC decoding, the $\text{SNR}_{dB}$ is usually assumed to be known and fixed, and the burst noise is often unexpected and its parameters are unknown to the decoder. Thus for learning based methods and traditional LDPC decoding method, the noisy signal $\tilde{\mathbf{y}}$ and the $\text{SNR}_{dB}$ are provided as input. In our experiments, the baselines includes bits decoding, sum-product based LDPC decoding and the Message Passing Neural Networks (MPNN).

**Architecture and training details** In this task, we use a factor graph neural network consisting of 8 FGNN layers (the details are provided in the supplementary file). The model is implemented using pytorch [27], trained with Adam optimizer [12] with initial learning

Table 2: Long-term prediction error (the smaller the better) of joint angles (top) and 3D joint positions (bottom) on H3.6M

| milliseconds | Walk 560 | 1000 | Eating 560 | 1000 | Smoking 560 | 1000 | Discussion 560 | 1000 | Average 560 | 1000 |
|---|---|---|---|---|---|---|---|---|---|---|
| convSeq2Seq[17] | N/A | 0.92 | N/A | 1.24 | N/A | 1.62 | N/A | 1.86 | N/A | 1.41 |
| GNN[20] | 0.65 | 0.67 | 0.76 | 1.12 | 0.87 | 1.57 | 1.33 | 1.70 | 0.90 | 1.27 |
| Ours | 0.67 | 0.70 | 0.76 | 1.12 | 0.88 | 1.57 | 1.35 | 1.70 | 0.91 | 1.27 |
| convSeq2Seq[17] | 69.2 | 81.5 | 71.8 | 91.4 | 50.3 | 85.2 | 101.0 | 143.0 | 73.1 | 100.3 |
| GNN[20] | 55.0 | 60.8 | 68.1 | 79.5 | 42.2 | 70.6 | 93.8 | 119.7 | 64.8 | 82.6 |
| Ours | 44.1 | 53.5 | 59.5 | 73.0 | 33.0 | 61.9 | 86.9 | 113.5 | 55.9 | 75.5 |

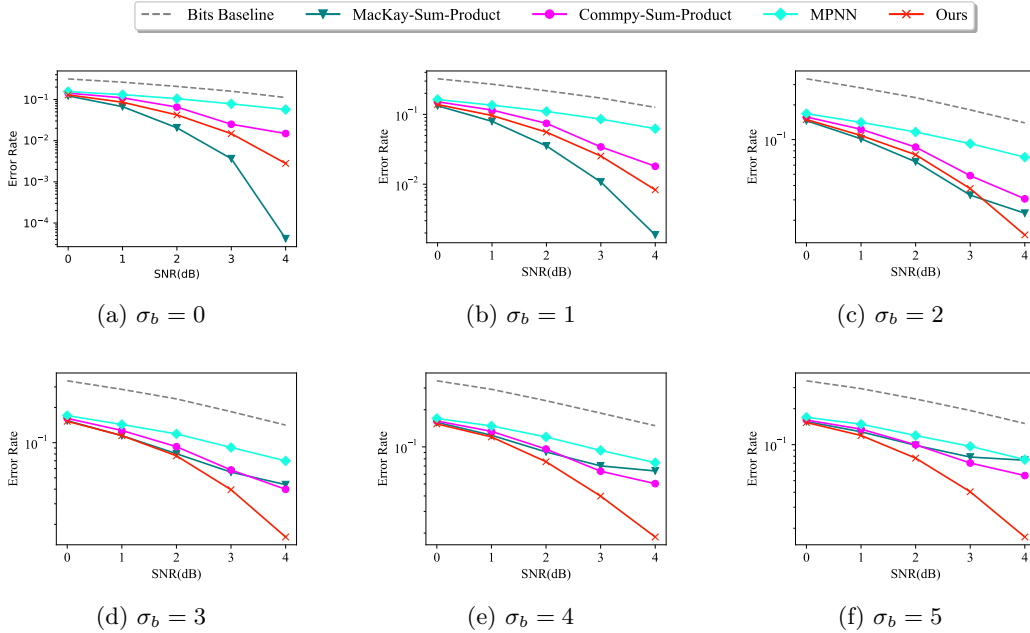

Figure 4: Experimental results on LDPC decoding.

rate lr $= 1 \times 10^{-2}$ and after every 10000 samples, lr is decreased by a factor of 0.98. After training over $10^8$ samples, the training loss converges. For MPNN, we use a 8 layer MPNN, and the same training protocol is used.

**Results**　We compare FGNN with two public available LDPC decoder MacKay-Sum-Product [19] and Commpy-Sum-Product [30]. Both the two decoder are using Sum-Product belief propagation to propagate information between higher order factors and nodes, but with different belief clipping strategy and different belief propagation scheduler. Meanwhile, our FGNN uses a learned factor-variable information propagation scheme, and the other learning based method, MPNN ignores the higher order dependencies. The decoding accuracy is provided in Figure 4. The MacKay-Sum-Product [19] is known to be near optimal for Gaussian noise and thus its performance is the best for lower burst noise level. The Commpy-Sum-Product have better performance than MacKay for high burst noise, but worse performance for low burst noise. Our FGNN always perform better than the Commpy-Sum-Product and MPNN, it achieves comparable but lower performance than the MacKay-Sum-Product for low burst noise level(0-2dB), and outperforms all other methods for high burst noise level(3-5dB).

## 4.3　Human Motion Prediction

The human motion prediction aims at predicting future motion of a human given a history motion sequence. As there are obviously higher order dependencies between joints, the factor graph neural network may help to improve the performance of the predictor. In this section,

we consider human motion prediction problem for the skeleton data, where the angle and 3d position of each joints are predicted. We build a factor graph neural network model for the skeleton data and compare the FGNN model with the state-of-the-art model based on GNN.

**Architecture and training details** We train our model on the Human3.6M dataset using the standard training-val-test split as previous works [17, 20, 22], and we train and evaluate our model using the same protocol as [20] (For details, see the supplementary file).

**Results** The results is provided in Table 2. For angle error, our FGNN model achieves similar results compared to the previous state-of-the-art GNN-based method [20], while for 3D position error, our model achieves superior performance. This is because compared to pairwise GNN, our model captures better higher order structural prior.

## 5    Conclusion

We extend graph neural networks to factor graph neural networks, enabling the network to capture higher order dependencies among the variables. The factor graph neural networks can represent the execution of the Max-Product algorithm on probabilistic graphical models, providing theoretical understanding on the representation power of graph neural networks. The factor graph provides a convenient method of capturing arbitrary dependencies in graphs and hypergraphs, including those with typed or conditioned nodes and edges, opening up new opportunities for adding structural bias into learning and inference problems.

## Broader Impact

Our work on the factor graph neural networks aims to make it easier to effectively specify structural inductive biases in the form of dependencies among sets of variables. This will impact on learning algorithms on structured data, particularly graph structured data. On the positive side, with well specified inductive biases, more effective learning would be possible on applications that require structured data. These include data with physical constraints such as human motion data, as well as data with abstract relationships such as social network data. On the negative side, in applications on some types of data such as social network data, better inference could mean less privacy. Research, guidelines, and possibly regulations on privacy can help to mitigate the negative effects.

## Acknowledgements

This work was supported by the National Research Foundation Singapore under its AI Singapore Program (Award Number: AISGRP- 2018-006). Any opinions, findings and conclusions or recommendations expressed in this material are those of the author(s) and do not reflect the views of National Research Foundation, Singapore. Zhen Zhang's participation was partially supported by the Australian Research Council Grant DP160100703.

## Footnotes

[1] Additional experiment on tree can be found in Appendix B.3 along with details on all experiments.

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
