[Supplementary Material]

# Factor Graph Neural Net—Supplementary File

## A  Proof of propositions

First we provide Lemma 8, which will be used in the proof of Proposition 2 and 4.

**Lemma 8.** *Given $n$ non-negative feature vectors $\mathbf{f}_i = [f_{i0}, f_{i1}, \ldots, f_{im}]$, where $i = 1, \ldots, n$, there exists $n$ matrices $\mathbf{Q}_i$ with shape $nm \times m$ and $n$ vector $\hat{\mathbf{f}}_i = \mathbf{Q}_i \mathbf{f}_i^T$, s.t.*

$$, \qquad [\mathbf{f}_1, \mathbf{f}_2, \ldots, \mathbf{f}_n] = [\max_i \hat{f}_{i0}, \max_i \hat{f}_{i1}, \ldots, \max_i \hat{f}_{i,mn}].$$

*Proof.* Let

$$\mathbf{Q}_i = \left[ \underbrace{\mathbf{0}^{m \times m}, \ldots, \mathbf{0}^{m \times m}}_{i-1 \text{ matrices}}, \mathbf{I}, \underbrace{\mathbf{0}^{m \times m}, \ldots, \mathbf{0}^{m \times m}}_{n-i \text{ matrices}} \right]^\top , \tag{8}$$

then we have that

$$\hat{\mathbf{f}}_i = \mathbf{Q}_i \mathbf{f}_i^T = \left[ \underbrace{0, \ldots, 0}_{(i-1)m \text{ zeros}}, f_{i0}, f_{i1}, \ldots, f_{im}, \underbrace{0, \ldots, 0}_{(n-i)m \text{ zeros}} \right]^\top .$$

By the fact that all feature vectors are non-negative, obviously we have that $[\mathbf{f}_1, \mathbf{f}_2, \ldots, \mathbf{f}_n] = [\max_i \hat{f}_{i0}, \max_i \hat{f}_{i1}, \ldots, \max_i \hat{f}_{i,mn}]$. $\qquad\square$

Lemma (8) suggests that for a group of feature vectors, we can use the $\mathcal{Q}$ operator to produce several $\mathbf{Q}$ matrices to map different vector to different sub-spaces of a high-dimensional spaces, and then our maximization aggregation can sufficiently gather information from the feature groups.

**Proposition 2.** *A factor graph $\mathcal{G} = (\mathcal{V}, \mathcal{C}, \mathcal{E})$ with variable log potentials $\theta_i(x_i)$ and factor log potentials $\varphi_c(\mathbf{x}_c)$ can be converted to a factor graph $\mathcal{G}'$ with the same variable potentials and the decomposed log-potentials $\varphi_{ic}(x_i, z_c)$ using a one-layer FGNN.*

*Proof.* Without loss of generality, we assume that $\log \phi_c(\mathbf{x}_c) \geqslant 1$. Then let

$$\theta_{ic}(x_i, z_c) = \begin{cases} \frac{1}{|s(c)|} \log \phi_c(\mathbf{x}_c^{z_c}), & \text{if } \hat{x}_i = x_i^{z_c}, \\ -c_{x_i, z_c}, & \text{otherwise,} \end{cases} \tag{9}$$

where $c_{x_i, z_c}$ can be arbitrary real number which is larger than $\max_{\mathbf{x}_c} \theta_c(\mathbf{x}_c)$. Obviously we will have

$$\log \phi_c(\mathbf{x}_c) = \max_{z_c} \sum_{i \in s(c)} \theta_{ic}(x_i, z_c) \tag{10}$$

Assume that we have a factor $c = 1, 2, \ldots n$, and each nodes can take $k$ states. Then $\mathbf{x}_c$ can be sorted as

$$\begin{aligned} [\,\mathbf{x}_c^0 &= [x_1 = 0, x_2 = 0, \ldots, x_n = 0], \\ \mathbf{x}_c^1 &= [x_1 = 1, x_2 = 0, \ldots, x_n = 0], \\ &\cdots, \\ \mathbf{x}_c^{k^n - 1} &= [x_1 = k, x_2 = k, \ldots, x_n = k]], \end{aligned}$$

and the higher order potential can be organized as vector $\mathbf{g}_c = [\log \phi_c(\mathbf{x}_c^0), \log \phi_c(\mathbf{x}_c^1), \ldots, \log \phi_c(\mathbf{x}_c^{k^n - 1})]$. Then for each $i$ the item $\theta_{ic}(x_i, z_c)$ in (9) have $k^{n+1}$ entries, and each entry is either a scaled entry of the vector $\mathbf{g}_c$ or arbitrary negative number less than $\max_{\mathbf{x}_c} \theta_c(\mathbf{x}_c)$.

Thus if we organize $\theta_{ic}(x_i, z_c)$ as a length-$k^{n+1}$ vector $\mathbf{f}_{ic}$, then we define a $k^{n+1} \times k^n$ matrix $\mathbf{Q}_{ci}$, where if and only if the $l^{\text{th}}$ entry of $\mathbf{f}_{ic}$ is set to the $m^{\text{th}}$ entry of $\mathbf{g}_c$ multiplied by

$1/|s(c)|$, the entry of $\mathbf{Q}_{ci}$ in $l^{\text{th}}$ row, $m^{\text{th}}$ column will be set to $1/|s(c)|$; all the other entries of $\mathbf{Q}_{ci}$ is set to some negative number smaller than $-\max_{\mathbf{x}_c} \theta_c(\mathbf{x}_c)$. Due to the assumption that $\log \phi_c(\mathbf{x}_c) \geqslant 1$, the matrix multiplication $\mathbf{Q}_{ci}\,\mathbf{g}_c$ must produce a legal $\theta_{ic}(x_i, z_c)$.

If we directly define a $\mathcal{Q}$-network which produces the above matrices $\mathbf{Q}_{ci}$, then in the aggregating part of our network there might be information loss. However, by Lemma 8 there must exists a group of $\tilde{\mathbf{Q}}_{ci}$ such that the maximization aggregation over features $\tilde{\mathbf{Q}}_{ci}\,\mathbf{Q}_{ci}\,\mathbf{g}_c$ will produce exactly a vector representation of $\theta_{ic}(x_i, z_c), i \in s(c)$. Thus if every $t_{ci}$ is a different one-hot vector, we can easily using one single linear layer $\mathcal{Q}$-network to produce all $\tilde{\mathbf{Q}}_{ci}\,\mathbf{Q}_{ci}$, and with a $\mathcal{M}$-network which always output factor feature, we are able to output a vector representation of $\theta_{ic}(x_i, z_c), i \in s(c)$ at each factor node $c$. $\qquad\square$

Given the log potentials represented as a set of rank-1 tensors at each factor node, we need to show that each iteration of the Max Product message passing update can be represented by a Variable-to-Factor layer followed by a Factor-to-Variable layer (forming a FGNN layer). We reproduce the update equations here.

$$b_{c \to i}(z_c) = \sum_{i' \in s(c), i' \neq i} \max_{x_i'} \left[ \log \phi_{i'c}(x_{i'}, z_c) + b_{i'}(x_{i'}) \right], \tag{11a}$$

$$b_i(x_i) = \theta_i(x_i) + \sum_{c: i \in s(c)} \max_z [\log \phi_{ic}(x_i, z_c) + b_{c \to i}(z_c)]. \tag{11b}$$

In the max-product updating procedure, we should keep all the decomposed $\log \phi_{i'c}(x_{i'}, z_c)$ and all the unary potential $\theta_i(x_i)$ for use at the next layer. That requires the FGNN to have the ability to fit the identity mapping. Consider letting the $\mathcal{Q}$ network to always output identity matrix, $\mathcal{M}([\mathbf{g}_c, f_i]|\Theta_{\text{VF}})$ to always output $\mathbf{g}_c$, and $\mathcal{M}([\mathbf{g}_c, f_i]|\Theta_{\text{FV}})$ to always output $f_i$. Then the FGNN will be an identity mapping. As $\mathcal{Q}$ always output a matrix and $\mathcal{M}$ output a vector, we can use part of their blocks as the identity mapping to keep $\log \phi_{i'c}(x_{i'}, z_c)$ and $\theta_i(x_i)$. The other blocks are used to updating $b_{c \to i}(z_c)$ and $b_i(x_i)$.

First we show that $\mathcal{M}$ operators in the Variable-to-Factor layer can be used to construct the computational graph for the max-marginal operations.

**Proposition 3.** *For arbitrary real valued feature matrix $\mathbf{X} \in \mathbb{R}^{m \times n}$ with $x_{ij}$ as its entry in the $i^{th}$ row and $j^{th}$ column, the feature mapping operation $\hat{\mathbf{x}} = [\max_j x_{ij}]_{i=1}^m$ can be exactly parameterized with a $2\log_2 n$-layer neural network with Relu as activation function and at most $2n$ hidden units.*

*Proof.* Without loss of generality we assume that $m = 1$, and then we use $x_i$ to denote $x_{1i}$. When $n = 2$, it is obvious that

$$\max(x_1, x_2) = \textbf{Relu}(x_1 - x_2) + x_2 = \textbf{Relu}(x_1 - x_2) + \textbf{Relu}(x_2) - \textbf{Relu}(-x_2)$$

and the maximization can be parameterized by a two layer neural network with 3 hidden units, which satisfied the proposition.

Assume that when $n = 2^k$, the proposition is satisfied. Then for $n = 2^{k+1}$, we can find $\max(x_1, \ldots, x_{2^k})$ and $\max(x_{2^k+1}, \ldots, x_{2^{k+1}})$ using two network with $2k$ layers and at most $2^{k+1}$ hidden units. Stacking the two neural network together would results in a network with $2k$ layers and at most $2^{k+2}$. Then we can add another 2 layer network with 3 hidden units to find $\max(\max(x_1, \ldots, x_{2^k}), \max(x_{2^k+1}, \ldots, x_{2^{k+1}}))$. Thus by mathematical induction the proposition is proved. $\qquad\square$

The update equations contain summations of columns of a matrix after the max-marginal operations. However, the VF and FV layers use max operators to aggregate features produced by $\mathcal{M}$ and $\mathcal{Q}$ operator. Assume that the $\mathcal{M}$ operator has produced the max-marginals, then we use the $\mathcal{Q}$ to produce several weight matrix. The max-marginals are multiplied by the weight matrices to produce new feature vectors, and the maximization aggregating function are used to aggregating information from the new feature vectors. We use the following propagation to show that the summations of max-marginals can be implemented by one MPNN layer plus one linear layer. Thus we can use the VF layer plus a linear layer to produce $b_{c \to i}(z_c)$ and use the FV layer plus another linear layer to produce $b_i(x_i)$. Hence to do $k$ iterations of Max Product, we need $k$ FGNN layers followed by a linear layer.

**Proposition 4.** *For arbitrary non-negative valued feature matrix $\mathbf{X} \in \mathbb{R}_{\geqslant 0}^{m \times n}$ with $x_{ij}$ as its entry in the $i^{th}$ row and $j^{th}$ column, there exists a constant tensor $\mathbf{W} \in \mathbb{R}^{m \times n \times mn}$ that can be used to transform $\mathbf{X}$ into an intermediate representation $y_{ik} = \sum_{ij} x_{ij} w_{ijk}$, such that after maximization operations are done to obtain $\hat{y}_k = \max_i y_{ik}$, we can use another constant matrix $\mathbf{Q} \in \mathbb{R}^{n \times mn}$ to obtain*

$$[\sum_i x_{ij}]_{j=1}^n = \mathbf{Q}[\hat{y}_k]_{k=1}^{mn}. \tag{12}$$

*Proof.* The proposition is a simple corollary of Lemma 8. The tensor $\mathbf{W}$ serves as the same role as the matrices $\mathbf{Q}_i$ in Lemma 8, which can convert the feature matrix $\mathbf{X}$ as a vector, then a simple linear operator can be used to produce the sum of rows of $\mathbf{X}$, which completes the proof. $\qquad\square$

In Lemma 8 and Proposition 4, only non-negative features are considered, while in log-potentials, there can be negative entries. However, for the MAP inference problem in (2), the transformation as follows would make the log-potentials non-negative without changing the final MAP assignment,

$$\tilde{\theta}_i(x_i) = \theta_i(x_i) - \min_{x_i} \theta_i(x_i), \qquad \tilde{\theta}_c(\mathbf{x}_c) = \theta_c(\mathbf{x}_c) - \min_{\mathbf{x}_c} \theta_c(\mathbf{x}_c). \tag{13}$$

As a result, for arbitary PGM we can first apply the above transformation to make the log-potentials non-negative, and then our FGNN can exactly do Max-Product Belief Propagation on the transformed non-negative log-potentials.

### A.1 A Factor Graph Neural Network Module Recovering the Belief Propagation

In this section, we give the proofs of Proposition 5 and 6 by constructing two FGNN layers which exactly recover the belief propagation operation. As lower order factors can always shrank by higher order factors, we will construct the FGNN layers on an factor graph $\mathcal{H} = (\mathcal{V}, \mathcal{F}, \hat{\mathcal{E}})$, which satisfies the following condition

1. $\forall i \in \mathcal{V}$, the associated $\theta_i(x_i)$ satisfies that $\theta_i(x_i) > 0 \forall x_i \in X$;

2. $\forall f_1, f_2 \in \mathcal{F}, |f_1| = |f_2|$;

3. $\forall f \in \mathcal{F}$, the corresponding $\varphi_f(\mathbf{x}_f)$ can be decomposed as

$$\varphi_f(\mathbf{x}_f) = \max_{z_f \in \mathcal{Z}} \sum_{i \in f} \varphi_{fi}(x_i, z_f), \tag{14}$$

and $\forall i \in f, \varphi_{fi}(x_i, z_f)$ satisfies that $\varphi_{fi}(x_i, z_f) > 0$.

On factor graph $\mathcal{H}$, we construct a FGNN layer on the directed bipartite graph in Figure 5.

Figure 5: Directed bipartite graph for constructing FGNN layers. In the Variable-to-Factor sub-graph, each factor receives the messages from the same number of nodes. On the other hand, for each Factor-to-Variable sub-graph, each nodes may receives messages from different number of factors.

**FGNN Layer to recover** (7a)    Here we construct an FGNN layer to produce all $b_{f \to i}(z_f)$. First we reformulate (7a) as

$$b_{f \to i}(z_f) = \tilde{\varphi}_f(z_f) - \max_{x_i}[\varphi_{if}(x_i, z_f) + b_i(x_i)],$$

$$\tilde{\varphi}_c(z_f) = \sum_{i \in f} \max_{x_i}[\varphi_{if}(x_i, z_f) + b_i(x_i)]. \tag{15}$$

Here we use the Variable-to-Factor sub-graph to implement (15). For each variable node $i$, we associated it with an length-$|X|$ vector $[b_i(x_i)]_{x \in X}$ (Initially $b_i(x_i) = \theta_i(x_i)$). For each edge in the sub-graph, assume that $f = [i_1, i_2, \ldots, i_{|f|}]$, then for some $i_j \in f$, the associated feature vector is as length-$|f|$ one-hot vector as follows

$$[0, 0, \ldots, \underbrace{1}_{\text{The } j^{\text{th}} \text{ entry.}}, \ldots, 0].$$

For each factor node $f = [i_1, i_2, \ldots, i_{|f|}]$ in the sub-graph, it is associated with an $|f| \times |X||Z|$ feature matrix as follows

$$\begin{bmatrix} [\varphi_{fi}(x_{i_1}, z_f)]_{x_{i_1}=1, z_f=1}^{x_{i_1}=|X|, z_f=|Z|} \\ [\varphi_{fi}(x_{i_2}, z_f)]_{x_{i_2}=1, z_f=1}^{x_{i_2}=|X|, z_f=|Z|} \\ \ldots \\ [\varphi_{fi}(x_{i_{|f|}}, z_f)]_{x_{i_{|f|}}=1, z_f=1}^{x_{i_{|f|}}=|X|, z_f=|Z|} \end{bmatrix}.$$

Then we construct an MPNN

$$\tilde{\mathbf{f}}_i = \max_{i \in f} \mathcal{Q}(\mathbf{e}_{f \to i}) \, \mathcal{M}(\mathbf{f}_i, \mathbf{f}_f), \tag{16}$$

as follows. The $\mathcal{Q}(\mathbf{e}_{f \to i})$ is an identity mapping. The $\mathcal{M}(\mathbf{f}_i, \mathbf{f}_f)$ consists of $|f|$ addition networks, where the $i_j^{\text{th}}$ networks will have an $|f| \times |X||Z|$ parameter

$$\begin{bmatrix} -\infty \\ -\infty \\ \ldots \\ [\varphi_{fi}(x_{i_j}, z_f)]_{x_{i_j}=1, z_f=1}^{x_{i_j}=|X|, z_f=|Z|} \\ \ldots \\ -\infty \end{bmatrix}.$$

In the $\mathcal{M}$-network, the $|f| \times |X||Z|$ parameter will be added to the $|f| \times |X||Z|$ and then the result will be reshaped to an $|f| \times |X| \times |Z|$ tensor. After that the tensor will be added to the length-$|X|$ feature vector of each nodes (reshaped to $1 \times 1 \times |X| \times 1$ tensor). In that case, for each $i_j \in f$, the $i_k^{\text{th}}$ will produce

$$\begin{bmatrix} -\infty \\ -\infty \\ \ldots \\ [\varphi_{fi}(x_{i_k}, z_f) + b_{i_j}(x_{i_j})]_{x_{i_k}=x_{i_j}=1, z_f=1}^{x_{i_k}=x_{i_j}=|X|, z_f=|Z|} \\ \ldots \\ -\infty \end{bmatrix}.$$

The $|f|$ $|f| \times |X| \times |Z|$ tensors will be stacked into an $|f| \times |f| \times |X| \times |Z|$ tensor, and it will be multiplied by the length-$|f|$ one-hot edge feature vector. That will produce

$$\begin{bmatrix} -\infty \\ -\infty \\ \ldots \\ [\varphi_{fi}(x_{i_j}, z_f) + b_{i_j}(x_{i_j})]_{x_{i_j}=1, z_f=1}^{x_{i_j}=|X|, z_f=|Z|} \\ \ldots \\ -\infty \end{bmatrix}.$$

Then the max operation over all $i \in f$ will produce edge feature matrix

$$
\begin{bmatrix}
[\varphi_{f i_1}(x_{i_1}, z_f) + b_{i_1}(x_{i_1})]_{x_{i_1}=1, z_f=1}^{x_{i_1}=|X|, z_f=|Z|} \\
[\varphi_{f i_2}(x_{i_2}, z_f) + b_{i_2}(x_{i_2})]_{x_{i_2}=1, z_f=1}^{x_{i_2}=|X|, z_f=|Z|} \\
\dots \\
[\varphi_{f i_{|f|}}(x_{i_2}, z_f) + b_{i_{|f|}}(x_{i_{|f|}})]_{x_{i_{|f|}}=1, z_f=1}^{x_{i_{|f|}}=|X|, z_f=|Z|}
\end{bmatrix}.
$$

Then by Proposition 3, we can recover the maximization operation in (15) using an $\mathcal{O}(\log_2|X|)$-layer neural network with at most $\mathcal{O}(|X|^2 \log_2|X|)$ hidden units. After that, all the other operations are simple linear operations, and they can be easily encoded in a neural-network without adding any parameter. Thus we can construct an FGNN layer, which produces factor features for each factor $f$ as follows

$$
\begin{bmatrix}
[b_{f \to i_1}(z_f)]_{z_f=1}^{z_f=|Z|} \\
[b_{f \to i_2}(z_f)]_{z_f=1}^{z_f=|Z|} \\
\dots \\
[b_{f \to i_{|f|}}(z_f)]_{z_f=1}^{z_f=|Z|}
\end{bmatrix}.
$$

Finally we constructed an FGNN to parameterize the operation in (7a), and this construction also proves Proposition 5 as follows.

**Proposition 5.** *The operation in* (7a) *can be parameterized by one MPNN layer with* $\mathcal{O}(|X| \max_{c \in \mathcal{C}} |\mathcal{Z}_c|)$ *hidden units followed by a* $\mathcal{O}(\log_2|X|)$-*layer neural network with at most* $\mathcal{O}(|X|^2 \log_2|X|)$ *hidden units.*

**FGNN Layer to recover** (7b)   Here we construct an FGNN layer to parameterize (7b) in order to prove Proposition 6. Using the notation in this section the operation in (7b) can be reformulated as

$$
b_i(x_i) = \theta_i(x_i) + \sum_{f: i \in f} \max_z [\varphi_{if}(x_i, z_f) + b_{c \to i}(z_f)].
$$

In previous paragraph, the new factor feature

$$
\begin{bmatrix}
[b_{f \to i_1}(z_f)]_{z_f=1}^{z_f=|Z|} \\
[b_{f \to i_2}(z_f)]_{z_f=1}^{z_f=|Z|} \\
\dots \\
[b_{f \to i_{|f|}}(z_f)]_{z_f=1}^{z_f=|Z|}
\end{bmatrix}.
$$

Considering the old factor feature

$$
\begin{bmatrix}
[\varphi_{f i}(x_{i_1}, z_f)]_{x_{i_1}=1, z_f=1}^{x_{i_1}=|X|, z_f=|Z|} \\
[\varphi_{f i}(x_{i_2}, z_f)]_{x_{i_2}=1, z_f=1}^{x_{i_2}=|X|, z_f=|Z|} \\
\dots \\
[\varphi_{f i}(x_{i_{|f|}}, z_f)]_{x_{i_{|f|}}=1, z_f=1}^{x_{i_{|f|}}=|X|, z_f=|Z|}
\end{bmatrix},
$$

we can use *broadcasted* addition between these two features to get

$$
\begin{bmatrix}
[b_{f \to i_1}(z_f) + \varphi_{f i}(x_{i_1}, z_f)]_{x_{i_1}=1, z_f=1}^{x_{i_1}=|X|, z_f=|Z|} \\
[b_{f \to i_2}(z_f) + \varphi_{f i}(x_{i_2}, z_f)]_{x_{i_2}=1, z_f=1}^{x_{i_2}=|X|, z_f=|Z|} \\
\dots \\
[b_{f \to i_{|f|}}(z_f) + \varphi_{f i}(x_{i_{|f|}}, z_f)]_{x_{i_{|f|}}=1, z_f=1}^{x_{i_{|f|}}=|X|, z_f=|Z|}
\end{bmatrix}.
$$

After that we have an $|f| \times |X| \times |Z|$ feature tensor for each factor $f \in \mathcal{F}$. By 3, a $\mathcal{O}(\log_2|\mathcal{Z}|)$-layer neural network with at most $\mathcal{O}(|\mathcal{Z}|^2 \log_2|\mathcal{Z}|)$ parameters can be used to convert the

above feature to

$$
\begin{bmatrix}
[\max_{z_f}[b_{f\to i_1}(z_f) + \varphi_{fi}(x_{i_1}, z_f)]]_{x_{i_1}=1}^{x_{i_1}=|X|} \\
[\max_{z_f}[b_{f\to i_2}(z_f) + \varphi_{fi}(x_{i_2}, z_f)]]_{x_{i_2}=1}^{x_{i_2}=|X|} \\
\cdots \\
[\max_{z_f}[b_{f\to i_{|f|}}(z_f) + \varphi_{fi}(x_{i_{|f|}}, z_f)]]_{x_{i_{|f|}}=1}^{x_{i_{|f|}}=|X|}
\end{bmatrix}.
$$

We will use this as the first part of our $\mathcal{M}$ network. For the second part, as we need to parameterize the $\sum_{f:i\in f}\max_z[\varphi_{if}(x_i, z_f) + b_{c\to i}(z_f)]$ from feature $\max_z[\varphi_{if}(x_i, z_f) + b_{c\to i}(z_f)]$, by Proposition 4, it will require another linear layer with $\mathcal{O}(\max_{i\in\mathcal{V}}\deg(\text{i})^2|X|^2)$, where $\deg(i) = |\{f|f \in \mathcal{F}, i \in f\}|$. After that, the $\mathcal{Q}$ network can be a simple identity mapping, and the FGNN would produce feature $\sum_{f:i\in f}\max_z[\varphi_{if}(x_i, z_f) + b_{c\to i}(z_f)]$ for each node. Adding these feature with the initial node feature would results new node feature $b_i(x_i)$. Thus by constructing a FGNN layer to parameterize (7b) we complete the proof of Proposition 6.

## B   Experiments

### B.1   Additional Ablation Study

**Aggregation Function**   In the Message Passing Neural Network module, various aggregation function such as "max", "sum" or "average" can be used. In our implementation, we choose the "max" aggregation function because theoretically "max" is invariant to the duplication of a element in the set, while "sum" or "average" is not. In real applications such as human motion prediction, different factor may have different size, but for better parallelization we may need to pad all factor to the same size. In this case, we may simply duplicate a node in factor to do this. We replaced the "max" aggregation with "sum" aggregation in the LDPC experiment and typical result is shown in Figure 6, where both algorithm achieve almost the same performance.

Figure 6: Comparison of "sum" and "max" aggregation.

### B.2   Additional Information on MAP Inference over PGM

**Data**   We construct four datasets. All variables are binary. The instances start with a chain structure with unary potential on every node and pairwise potentials between consecutive nodes. A higher order potential is then imposed to every node for the first three datasets.

The node potentials are all randomly generated from the uniform distribution over $[0, 1]$. We use two kinds of pairwise potentials, one randomly generated (as in Table 4), the other encouraging two adjacent nodes to both take state 1 (as in Table 3 and Table 5), i.e. the potential function gives high value to configuration $(1, 1)$ and low value to all other configurations. For example, in Dataset1, the potential value for $x_1$ to take the state 0 and $x_2$ to take the state 1 is 0.2; in Dataset3, the potential value for $x_1$ and $x_2$ to take the state 1 at the same time is sampled from a uniform distribution over $[0, 2]$.

| pairwise potential | $x_2 = 0$ | $x_2 = 1$ |
|:---:|:---:|:---:|
| $x_1 = 0$ | 0 | 0.1 |
| $x_1 = 1$ | 0.2 | 1 |

Table 3: Pairwise Potential for Dataset1

| pairwise potential | $x_2 = 0$ | $x_2 = 1$ |
|:---:|:---:|:---:|
| $x_1 = 0$ | U[0,1] | U[0,1] |
| $x_1 = 1$ | U[0,1] | U[0,1] |

Table 4: Pairwise Potential for Dataset2,4

| pairwise potential | $x_2 = 0$ | $x_2 = 1$ |
|:---:|:---:|:---:|
| $x_1 = 0$ | 0 | 0 |
| $x_1 = 1$ | 0 | U[0,2] |

Table 5: Pairwise Potential for Dataset3

For Dataset1,2,3, we additionally add the budget higher order potential [23] at every node; these potentials allow at most $k$ of the 8 variables that are within their scope to take the state 1. For the first two datasets, the value $k$ is set to 5; for the third dataset, it is set to a random integer in {1,2,3,4,5,6,7,8}. For Dataset4, there is no higher order potential.

As a result of the constructions, different datasets have different inputs for the FGNN; for each dataset, the inputs for each instance are the parameters of the PGM that are not fixed. For Dataset1, only the node potentials are not fixed, hence each input instance is a factor graph with the randomly generated node potential added as the input node feature for each variable node. Dataset2 and Dataset4 are similar in terms of the input format, both including randomly generate node potentials as variable node features and randomly generated pairwise potential parameters as the corresponding pairwise factor node features. Finally, for Dataset3, the variable nodes, the pairwise factor nodes and the high order factor nodes all have corresponding input features.

**Architecture** We use a multi-layer factor graph neural network with architecture FGNN(64) - RES[FC(64) - FGNN(64) - FC(64)] - MLP(128) - RES[FC(64) - FGNN(64) - FC(128)] - FC(256) - RES[FC(256) - FGNN(64) - FC(256)] - FC(128) - RES[FC(128) - FGNN(64) - FC(128)] - FC(64) - RES[FC(64) - FGNN(64) - FC(64)] - FGNN(2). Here one FGNN($C_{\text{out}}$) is a FGNN layer with $C_{\text{out}}$ as output feature dimension with ReLU [26] as activation. One FC($C_{\text{out}}$) is a fully connected layer with $C_{\text{out}}$ as output feature dimension and ReLU as activation. Res[·] is a neural network with residual link from its input to output; these additional architecture components can assist learning.

**Running Time** We report the inference time of one instance and the training time of one epoch for the synthetic datasets in Table 6. The results show that our method runs in a reasonable amount of time.

| ($\mu$s) | PointNet | DGCNN | AD3 (exact/approx) | Max-Product | MPLP | MPNN | Ours |
|---|---|---|---|---|---|---|---|
| D1 | 45 (43) | 285 (107) | 5 / 5 | 6 | 57 | 131 (72) | 144 (75) |
| D2 | – | – | 532 / 325 | 1228 | 55 | 131 (72) | 341 (162) |
| D3 | – | – | 91092 / 1059 | 4041 | 55 | 121 (74) | 382 (170) |
| D4 | – | – | 6 / 5 | 6 | 0.04 | 137 (71) | 216 (101) |

Table 6: Inference time in microseconds of one instance on synthetic datasets and GPU training time of one epoch in milliseconds (in bracket) for applicable methods.

## B.3 Experiment on tree structured PGM

Apart from the chain structured PGM in Section 4.1, we also have an additional experiment on tree structured PGM. The training set includes 90000 different PGM as randomly generated binary trees whose depth are between 3 and 6. Each node is associated with a random variable $x_i \in \{0, 1\}$ along with a log potential $\theta_i(x_i)$ randomly sampled from

Gaussian distribution $\mathcal{N}(0,1)$. Each edge $(i,j)$ in the tree is associated with a pairwise log potential $\theta_{ij}(x_i, x_j)$ which is randomly sampled from Gaussian distribution $\mathcal{N}(0,1)$. There is also 10000 testing instances which is generated in the same way as the training set. The experiment result is shown in Table 7.

| | AD3 | Max-Product | MPLP | MPNN | Ours |
|---|---|---|---|---|---|
| Agreement on MAP | 1.0 | 1.0 | 0.9997 | – | 0.9835 |

Table 7: Experimental result on tree structured PGM.

For a tree structured PGM, it is not as easy to shrink the pairwise features to the nodes as an adaptation for MPNN as in the case of chain PGM in Section 4.1, so we omit the experiment on MPNN. Still, our Factor Graph Neural Network achieves a good performance even when compared with Max-Product which is optimal on tree PGMs and also with the linear programming relaxations.

### B.4 Testing on novel graph structures for synthetic data

We conducted a new experiment to train the FGNN on fixed length-30 MRFs using the same protocol as Dataset3, and test the algorithm on 60000 random generated chain MRF whose length ranges from 15 to 45 (the potentials are generated using the same protocol as Dataset3). The result is in Table 8, which shows that the model trained on fixed size MRF can be generalized to MRF with different graph structures.

| Chain length | AD3 | FGNN |
|---|---|---|
| (15, 25) | 88.95% | 94.31% |
| (25, 35) | 88.18% | 93.64% |
| (35, 45) | 87.98% | 91.50% |

Table 8: Accuracy on dataset with different chain size.

### B.5 Implementation details on MAP Solvers

In the experiment, the AD3 code is from the official code repo [3], which comes with a python interface. For Max-Product algorithm, we use the implementation from libdai and convert the budget higher potential as a table function. For the MPLP algorithm, we implemented it in C++ to directly support the budget higher order potential. The re-implemented version is compared with the original version [4], and its performance is better than the original one in our experiment. So we provide the result of the re-implemented version.

### B.6 Dataset Generation and Training Details of LDPC decoding

**Data**  Each instance of training/evaluation data is generated as follows:

During the training of MPNN and FGNN, the node feature include the noisy signal $\tilde{\mathbf{y}}$ and the signal-to-noise ratio $\mathrm{SNR}_{dB}$. For MPNN, no other feature are provided, while for FGNN, for each factor $f$, the vector $[\tilde{y}_i]_{i \in f}$ is provided as feature vector. Meanwhile, for each edge from factor node $f$ to one of its variable node $i$, the factor feature and the variable node feature are put together to get the edge feature.

**Architecture**  In our FGNN, every layer share the same $\mathcal{Q}$ network, which is 2-layer network as follows MLP(64)-MLP(4). Here the first layer comes with a ReLU activation function and the second layer is with no activation function.

**Algorithm 2** Data Generation for LDPC decoding

---

**Output: y**: a 96-bit noisy signal; $\mathrm{SNR}_{dB}$: signal-to-noise ratio, a scalar

Uniformly sample a 48-bit binary signal **x**, where for each $0 < i \leqslant 48$, $P(x_i = 1) = P(x_i = 0) = 0.5$

Encode **x** using the "96.3.963" scheme [19] to get a 96-bit signal **y**

sample $\mathrm{SNR}_{dB} \in \{0, 1, , 2, 3, 4\}$ and $\sigma_b \in \{0, 1, , 23, 4, 5\}$ uniformly

For each $0 < i \leqslant 96$, uniformly, sample

- $\eta_i \in \mathcal{U}(0, 1)$,
- $n_i \in \mathcal{N}(0, \sigma^2)$ s.t. $\mathrm{SNR}_{dB} = 20 \log_{10} 1/\sigma$
- $z_i \in \mathcal{N}(0, \sigma_b^2)$

Set noisy signal **ỹ** to

- $\tilde{y}_i = y_i + n_i + \mathbb{I}(\eta_i \leqslant 0.05)z_i$

---

The overall structure of our FGNN is as follows INPUT - RES[FC(64) - FGNN(64) - FC(64)] - RES[FC(64) - FGNN(64) - FC(64)] - FC(64) - FGNN(64) - FC(128) - FC(256) - FGNN(128) - FC(256) - - RES[FC(256) - FGNN(128) - FC(256)] - FC(128) - FGNN(128) - FC(128) - FC(64) - FGNN(64) - FC(64) - RES[FC(64) - FGNN(64) - FC(64)] - FC(128) - FC(128) - FC(1). In the network, a batch-normalization layer and a ReLU activation function is after each FC layer and FGNN layer except for the last FC layer.

## B.7   Details of Human Motion Prediction

For human motion prediction, we are using the Human 3.6M (H3.6M) dataset. In this experiment, we replace the last two GNN layer in Mao et al. [20]'s model with FGNN layer with the same number of output channels. The H3.6M dataset includes seven actors performing 15 varied activities such as walking, smoking *etc.*. The poses of the actors are represented as an exponential map of joints, and a special pre-processing of global translation and rotation. In our experiments, as in previous work[17, 20], we only predict the exponential map of joints. That is, for each joints, we need to predict a 3-dimensional feature vector. Thus we add a factor for the 3 variable for each joint [5]. Also for two adjacent joint, a factor of 6 variables are created. The factor node feature are created by put all its variable node feature together. For the edge feature, we simply use one hot vector to represent different factor-to-variable edge. For evaluation, we compared 4 commonly used action — walk, eating, smoking and discussion. The result of GNN and convSeq2Seq are taken from [20], and our FGNN model also strictly followed the training protocol of [20].

## Footnotes

[3]https://github.com/andre-martins/AD3

[4]https://people.csail.mit.edu/dsontag/code/mplp_ver2.tgz

[5]In practice, those angles with very small variance are ignored, and these variables are not added to the factor graph