[Reviews · NeurIPS 2020]

Review 1

Summary and Contributions: 1. the paper proposes a neural network on factor graphs for MAP inference; 2. the paper proves that the max-product algorithm is a special case of it (though there can be exponential number of rank-1 tensors); 3. evaluation on synthetic data, LDPC decoding, and human motion prediction.

Strengths: I think learnable inference algorithm for PGMs is definitely interesting and novel topic to research. This paper is novel by bridging graph neural network and max-product algorithms. Impressive empirical results are presented. The results in this paper can be further explored by the probabilistic graphical model community, to improve current inference algorithms.

Weaknesses: 1. I am not entirely sure how useful is the theoretical result in Sec. 3.2. First of all, the decomposition may result in exponential rank-1 tensors, does it mean the size of the network should be exponential? Secondly, I think the point of this work is a more powerful MAP inference algorithm than the max-product algorithm (as the empirical result shows). However, the theory does not explain why the proposed algorithm can be more powerful than max-product. 2. the synthetic data experiment might be unfair, since the neural network is trained with exact MAP solutions, and it can possibly over-fit the solution. Furthermore, only belief propagation algorithms are compared. How does variational inference work for these tasks?

Correctness: I have not check the theory thoroughly. According to my understanding, the claims and method seem reasonable. The empirical methodology generally sounds, but may have some flaws, as pointed out above.

Clarity: The writing needs to be improved. 1. there are many typos (missing spaces and citations); 2. Q(t | Phi), M([g, f] | Theta), Phi, and Theta in Algorithm 1 are undefined; 3. There are many lemmas and propositions in Sec. 3.2 as the sketch of the proof. However, I cannot find the main theorem itself in the main text.

Relation to Prior Work: It clearly discussed the difference from MPNN. I am not aware if there are any other works on learnable MAP inference algorithms though.

Reproducibility: Yes

Additional Feedback: Post rebuttal: I think my concerns about the theory and experiments are addressed in the rebuttal. Therefore I increase my score from 5 to 6.


Review 2

Summary and Contributions: The paper generalizes graph neural networks to factor graphs

Strengths: Relevant work to NeurIPS community, established connection with belief propagation

Weaknesses: Detailed motivation on the choice of the architecture, aggregation function, and some empirical details are lacking (see below)

Correctness: Yes

Clarity: Yes

Relation to Prior Work: Yes

Reproducibility: Yes

Additional Feedback: - Max-product is known not to decode well LDPC codes, that’s probably why the authors compared to schemes that use sum-product. But then why is max used in the architecture? Does the scheme work well for the sum rule instead of max? - The training time is not reported (50 epochs – how much time is that?), only inference time. Doesn’t this automatically represent a computational disadvantage of graph neural networks compared to message-passing methods? - The number of potential input sequences to LDPC code is huge, how representative is the test with 1000 instances? ---------------- Post-rebuttal: I am satisfied with the additional clarifications and evidence presented by the authors during the rebuttal. It would be beneficial to include these additional explanations in the final version of the paper. I think that overall this paper is a good study, and I am keeping my original score 7 (accept).


Review 3

Summary and Contributions: This paper proposes a message passing neural network that is theoretically guaranteed to express certain higher-order dependencies among the random variables. To be specific, the proposed parameterization generalizes the max-product belief propagation (BP) run on some graphical model with higher-order interactions. In the experiment, the proposed approach outperforms over several baselines.

Strengths: This paper proposes a new member of the message passing neural network (MPNN) with analyzable expressive power. The authors establish some theoretical analysis on the power of the proposed architecture (by showing it to generalize the max-product BP). Experiments were done across several tasks.

Weaknesses: The proposed architecture is not particularly novel and experiments can be improved. While the theoretical analysis is quite interesting, it is not significant enough to bypass the aforementioned issues (e.g., the analysis mainly relies on the Lemma 1 proposed by Kohli et al.). While the proposed factor graph neural network (FGNN) is guaranteed to express a family of higher-order interactions, in the end, FGNN is a member of MPNN applied to heterogeneous graph with two types of vertices (random variable and factor). I also think the considered experiments are limited since they only consider the case where (1) training and evaluation are done on the same graph and (2) factors are easily expressed as a representation of fixed dimension. In other words, the considered experiments are not very convincing for showing that the proposed FGNN works across general graphical models. Especially, I think it is critical to consider training FGNN on factor graphs where (2) is not satisfied, e.g., represented by (strictly positive) higher-order tensors with elements sampled from a uniform distribution. This would be useful for validating that FGNN is indeed able to model the higher-order interactions in general graphical models with higher-order interactions, as analyzed in Section 4.

Correctness: The proposed method and analysis are correct to my knowledge.

Clarity: I think the architecture of FGNN should be explained in more detail, e.g., explaining more about the right side of Figure 3 in the main text.

Relation to Prior Work: The authors should discuss and reference the existing work using graph neural network to perform inference in a graphical model (defined on pair-wise interactions), i.e., Yoon et al. 2018. Yoon et al. 2018, Inference in Probabilistic Graphical Models by Graph Neural Networks

Reproducibility: Yes

Additional Feedback: I think the paper should more clearly describe how the FGNN works on heterogeneous graphs with fixed-size representation for both the random variable and the factors in Section 3.1. Explicitly stating (in the main paper) on how the input representation is constructed from the higher-order factor graph would also help. I also suggest the authors to either publish the code used for the experiment, or provide a more detailed description of the experimental settings and the baselines considered. ============ I appreciate the authors' detailed and thoughtful rebuttal. However, I still think the experiment on higher-order factor graphs is necessary (since this is related to main theoretical result of the paper). I do not think the human motion prediction task is a good example in this case because it is more commonly expressed as a set of variables with pairwise interactions. Hence, I am keeping my current score.

[Author Response · NeurIPS 2020]

We would like to thank the reviewers for the insightful remarks and comments. We address several concerns of the reviewers as follows.

**Main theoretical results and practical implications: R1 and R3.** Our main theoretical result is that the Max-Product algorithm can be exactly parameterized by the FGNN using number of parameters that are polynomial w.r.t. number of rank-1 tensors required to represent the higher order potentials (Corollary 7 in the main text). In worst case, it may require exponentially many parameters for arbitrary higher order potentials. However, we believe that there will be multiple practical applications that have reasonably sized representation. This will have to be verified empirically – we have demonstrated this for error correcting codes which showed improved performance under bursty channel, and for human motion prediction which showed state-of-the-art performance. Furthermore, in computer vision applications, commonly used handcrafted higher order potentials, *e.g.* Potts model or Robust Potts Model [DOI:10.1007/s11263-008-0202-0], can often be exactly or approximately represented using a moderate number of rank-1 tensors. FGNN is also advantageous when the potential functions are unknown and only the features corresponding to the factor are provided. In this case, we can use FGNN to learn the approximation of potential functions from data, where the size of the network can be decided based on the computational budget or amount of available data (to control over-fitting). As FGNN can exactly represent the Max-Product, it should be at least as powerful as the Max-Product algorithm. It can represent a set of algorithms including the Max-Product, so by learning from data it may potentially learn a better inference algorithm if Max-Product is not optimal. In our experiment, we show that FGNN outperforms the Max-Product algorithm.

**Possible over-fitting for synthetic data: R1.** In the synthetic data experiment, our goal is to train a better MAP inference algorithm, and thus the MAP assignment is used as the target for all neural network based algorithms. In the experiment, the number of all possible configurations is $2^{30} \approx 10^9$ for a length-30 binary chain structured MRF, and our training set only has 90000 items. In this case, it is unlikely that a neural network that simply over-fits the training set can have a good performance on the test set.

**Testing on novel graph structures: R3 and R1.** To address R3's concern that the algorithm is tested only on the same graph structure as seen in training, we conducted a new experiment to train the FGNN on fixed length-30 MRFs using the same protocol as Dataset3, and test the algorithm on 60000 random generated chain MRF whose length ranges from 15 to 45 (the potentials are generated using the same protocol as Dataset3). The result is in Table 1, which shows that the model trained on fixed size MRF can be generalized to MRF with different graph structures. This also further addresses the overfitting issue raised by R1.

| Chain length | AD3 | FGNN |
|---|---|---|
| (15, 25) | 88.95% | 94.31% |
| (25, 35) | 88.18% | 93.64% |
| (35, 45) | 87.98% | 91.50% |

Table 1: Accuracy on dataset with different chain size.

**Factor without trivial representation of fixed dimension: R3.** In the experiment on human motion prediction, the potential functions are actually unknown and their representations are fully learned from data.

**Variational inference: R1.** The synthetic problems are MAP inference problems. As variation inference is designed for marginal inference, it cannot be directly used. Indirectly however, by using the zero-temperature technique, the variational inference can be applied to do MAP inference. By applying the zero-temperature technique, the objective function of the variational inference becomes equivalent to that of AD3 and MPLP. Particularly, the AD3 algorithm is guaranteed to converge to the optimal of that objective and thus it can be viewed as a variant of variational inference.

**Improve the writing: R1.** We will revise Section 3 to say that Corollary 7 is the main theoretical results of the paper and do careful proofreading. For Algorithm 1, $\Phi_{VF}$ and $\Theta_{VF}$ are parameters of the $\mathcal{Q}$ and $\mathcal{M}$ net in the Variable to Factor module, respectively. $\Phi_{FV}$ and $\Theta_{FV}$ are parameters of the $\mathcal{Q}$ and $\mathcal{M}$ net in the Factor to Variable module.

**Aggregation function: R2.** We choose the "max" aggregation function because theoretically "max" is invariant to the duplication of a element in the set, while "sum" or "average" is not. In real applications such as human motion prediction, different factor may have different size, but for better parallelization we may need to pad all factor to the same size. In this case, we may simply duplicate a node in factor to do this. We replaced the "max" aggregation with "sum" aggregation in the LDPC experiment and typical result is shown in Figure 1, where both algorithm achieve almost the same performance.

Figure 1: Comparison of "sum" and "max" aggregation.

**Training time: R2.** The training time is in Table 6 of the supplementary file.

**Representativeness of the LDPC data: R2.** In this experiment, as previous work [Kim et al., 2018, Zarkeshvari and Banihashemi, 2002] we do not have any assumption on the input signal, thus both the train and test set the input signal are uniformly sampled from $\{0, 1\}^{48}$. We can add error bars in the next version to indicate how well the results represent uniformly distributed inputs.

**Relation with Yoon *et al.* and publishing code: R3.** We will add discussion on relation with Yoon *et al.*, which is on pairwise potentials, as well as detailed experiment settings and the url to the source code in the next version.

[Meta-Review · NeurIPS 2020]

This paper generalizes graph neural networks to factor graphs, and shows that the max-product algorithm is a special case (though there can be exponential number of rank-1 tensors). Experiments synthetic data, LDPC decoding, and human motion prediction are presented. The major concerns were addressed in the rebuttal, which provides additional clarifications and evidence. The authors should include them in the final version of the paper.